

# What controls the seasonal cycle of columnar methane observed by GOSAT over different regions in India?

Naveen Chandra[1*], Sachiko Hayashida[1], Tazu Saeki[2], and Prabir K. Patra[2]

[1] Nara Women's University, Kita-Uoya Nishimachi, Nara 630-8506, Japan

[2] Department of Environmental Geochemical Cycle Research, JAMSTEC, Yokohama 2360001, Japan

*Correspondence to*: Naveen Chandra (nav.phy09@gmail.com)

**Abstract.** Methane ($CH_4$) is one of the most important short-lived climate forcer (SLCFs) according to the United Nations Environment Programme as well as it plays also a critical role in air pollution chemistry in the troposphere. With the availability of satellite observations from space, variabilities in $CH_4$ have been captured for most parts of the global land with major emissions. The satellite observations however do not allow us to derive emission information straightforwardly, unlike in-situ measurements near the source region, without separating the role of transport and chemistry in the columnar

dry-air mole fractions of methane ($XCH_4$), which involves the $CH_4$ densities at all altitudes along the solar light path. Observations of enhanced $XCH_4$ are often linked to the local/regional emissions by simple correlation, without separating transport and chemistry contributions to $XCH_4$ variability. Here, we present the analysis of $XCH_4$ variability over different inland and surrounding cleaner oceanic regions of India using GHGs Observation SATellite (GOSAT). We also use the JAMSTEC's state-of-the-art atmospheric chemistry-transport model (ACTM) for simulating the observed $XCH_4$

concentrations by varying surface emissions. The model-observation comparisons help us to elucidate the synoptic and seasonal variabilities in $XCH_4$ in relation with coupled monsoon meteorology and surface fluxes. Distinct seasonal variations of $XCH_4$ have been observed over the regions lying in the northern (north of $15^oN$) and southern part (south of $15^oN$) of India, corresponding to the peak during southwest (SW) monsoon (July-September) and early autumn season (October-December), respectively. The detailed study of all possible governing factors (transport, emission and chemistry) responsible

for $XCH_4$ seasonal cycle suggests that distinct $XCH_4$ seasonal cycle over northern and southern regions of India is not only governed by the heterogeneous distributions of surface emissions, but also distribution of $CH_4$ in the upper tropospheric layer. We have observed different contributions from lower troposphere (~1000-600 hPa), affected mainly by surface emissions, and transport dominated upper atmosphere (~600-0 hPa) in the $XCH_4$ seasonal cycle. Over most of northern part of the Indian regions, up to 40% of the peak during the SW monsoon season is attributed to the lower troposphere, while

~60% to uplifted high-$CH_4$ air masses in the upper atmosphere. In contrast, $XCH_4$ enhancement over the semi-arid western India is mainly (~88%) attributed to the upper atmosphere. The ratios of contribution changed over the southern peninsula and cleaner oceanic region; up to 60% of seasonal cycle of $XCH_4$ is contributed by lower tropospheric region. These





differences arise due to the complex atmospheric transport mechanisms, caused by the seasonally varying monsoon. The $CH_4$ enriched air masses uplifted from high emission region by the SW monsoon circulation and deep cumulus convection,

and then confined by anticyclonic wind in the upper troposphere (~200 hPa), cause the strong contribution of the upper troposphere in the peak of $XCH_4$ over most of the regions lying in the northern part of India. Based on this analysis, we suggest that a link between surface emissions and higher levels of $XCH_4$ is not always valid over Asian monsoon regions, although there is often a fair correlation between surface emissions and $XCH_4$.

**1. Introduction**

Methane ($CH_4$) is the second most important anthropogenic greenhouse gas (GHG) after carbon dioxide ($CO_2$) and accounts for ~20% (+0.97 W m$^{-2}$) of the increase in total direct radiative forcing, since 1750 (Myhre et al., 2013). $CH_4$ is emitted from a range of anthropogenic and natural sources in the atmosphere. The main natural sources of $CH_4$ include wetlands and termites (Cao et al., 1998; Sugimoto et al., 1998). Livestock, rice cultivation, fossil fuel industry (production and uses of

natural gas, oil and coal) and landfills are the major sectors among the anthropogenic sources (Crutzen et al., 1986; Minami and Neue, 1994, Olivier et al., 2006; EDGAR2FT, 2013).

With a short atmospheric lifetime of about 10 years (e.g., Patra et al., 2011) and having 34 times more potential to trap heat than $CO_2$ on per molecule basis over a 100-year timescale, mitigation of $CH_4$ emissions could be the most important way to

limit global warming at inter-decadal time scales (Shindell et al., 2009). Better knowledge of $CH_4$ distribution and quantification of its emission flux is indispensable for assessing possible mitigation strategies. However, sources of $CH_4$ are not yet well quantified due to sparse ground based measurements, which results in limited representation on a larger scale (Dlugokencky et al., 2011; Patra et al., 2016). Recent technological advances have made it possible to detect spatial and temporal variations in atmospheric $CH_4$ from space (Frankenberg et al., 2005; Kuze et al., 2009), which could fill in gap left

by ground-based measurements, albeit at a lower accuracy than *in situ* measurements. Although satellite observations have the advantage of providing continuous monitoring over a wide spatial range, the information obtained from passive nadir-sensors, which use solar radiation at Short-Wavelength Infrared (SWIR) spectral region, is limited to $XCH_4$. This is an integrated measure of $CH_4$ with contributions from the different vertical atmospheric layers, i.e., from the Earth's surface to the top of the atmosphere (up to about 100km).


The Indian region exerts a significant impact on the global $CH_4$ emissions. The Indo-Gangetic Plane (IGP), mostly hot and humid northeast region lies in the foothills of the Himalayas, is one of the most polluted regions in the world due to the high population density, heavy industries, power plants that host 70% of coal-fired thermal power plants in India and intense agricultural activity (Kar et al., 2010). It is well known from previous studies that southwest (SW) monsoon (July-

September) meteorology affects significantly the pollutants including $CH_4$ mixing ratios from surface to upper troposphere (400 – 200 hPa) (Park et al., 2004; Randel et al., 2006; Xiong et al., 2009; Baker et al., 2012; Schuck et al., 2012; Lal et al.,





2014, Chandra et al., 2016). Rain during the SW monsoon season cause higher $CH_4$ emissions from the paddies field and wetlands (e.g., Hayashida et al., 2013) while the persistent deep convection uplift the $CH_4$-laden air mass from surface to the upper troposphere during the same season (Baker et al., 2012; Schuck et al., 2012), where $CH_4$ is further spread over larger

region by the anticyclonic winds. The dynamical system dominated by deep convection and anticyclone cover mostly the northern Indian region (north of $15^oN$) due to the presence of Himalayas and Tibetan Pleatu, while such complex dynamical system has not been observed over the southern part of India (south of $15^oN$) (Rao, 1976).

Satellite-based measurements show elevated levels of $XCH_4$ over the northern part of India (north of $15^oN$) particularly high

over IGP during the southwest (SW) monsoon season (July to September) and over southern India (south of $15^oN$) during early autumn (October to December) season (Frankenberg et al., 2005, 2006; Hayashida et al., 2013). Previous studies related the high $XCH_4$ values correspond to the strong surface $CH_4$ emissions particularly from the rice cultivation over the Indian region, because they showed statistically significant correlations over certain regions (Hayashida et al., 2013; Kavitha et al., 2016).  The differences in the peak of $XCH_4$ seasonal cycle over northern and southern regions of India are discussed

on the basis of agricultural practice in India that takes place in two seasons, namely the Kharif (May to October) and Rabi (November to April), respectively. However, inferring the local emissions to the higher levels of $XCH_4$, which involves all of the $CH_4$ abundances at any altitude along the solar light path, is highly ambiguous, particularly over the regions where monsoon meteorology significantly affects the tropospheric distributions of $CH_4$ by coupling the surface emission to upper tropospheric air. Therefore, data indicating the vertical distributions of $CH_4$ are important when measurements of $XCH_4$ are

used to investigate underlying surface emissions.

Although GOSAT has also a Thermal InfraRed (TIR) channel for providing the vertical distributions of $CH_4$, but the retrieval products are still under validation (e.g., Zou et al., 2016, Olsen et al., 2017). There are some other infrared sensors such as AIRS and TES, which could provide some information of vertical distributions of $CH_4$, but they are not sufficiently

sensitive to the lower troposphere (approximately bias ~ 65 ppb) (Worden et al., 2015). Some studies have been done to discriminate lower tropospheric $CH_4$ from the upper part, but most of work is still under validation (e.g., Worden et al., 2015). Under the limitations of satellite in providing information on vertical $CH_4$ distributions, the Atmospheric Chemistry transport model (ACTM) becomes a powerful tool to understand its vertical distribution along with major controlling factors emissions, transport, and chemistry processes separately. This manuscript attempts for the first time to separate the factors

responsible (emission, transport and chemistry) for the distributions of columnar methane ($XCH_4$) over the Asian monsoon region for different altitude segments. The $XCH_4$ mixing ratios, observed from the Greenhouse gases Observing SATellite (GOSAT) and simulated from JAMSTEC's ACTM, are used for this study. The main conclusion drawn from this study is that a link between surface emission seasonality and higher $XCH_4$ values over the Asian monsoon region is not always valid, although there is often a fair correlation between them. Transport dominated upper atmospheric layer contribute significantly

in the higher levels of XCH4 during the seasons of high convective activity.





## 2. Methods

### 2.1 Satellite data:

The Greenhouse gases Observing SATellite (GOSAT) (called Ibuki) is a joint satellite project of National Institute of Environmental Studies (NIES), Ministry of the Environment (MOE) and Japan Aerospace Exploration Agency (JAXA). It has been providing global observations of columnar mixing ratios of greenhouse gases ($XCH_4$ and $XCO_2$) since its launch in January 2009. It is equipped with onboard thermal and near infrared sensors for carbon observations (TANSO), which include a Fourier transform spectrometer (FTS) for GHG's monitoring and a cloud and aerosol imager (CAI) to detect cloud

and aerosols in the FTS field of view (Kuze et al., 2009). Cloudy data is strictly screened using simultaneously recorded atmospheric images from CAI. As a result of strict screening, only limited numbers of $XCH_4$ data are available during the SW monsoon season over South Asia. This study uses the GOSAT SWIR $XCH_4$ (Version 2.21)-Research Announcement product for the period of 2011-2014. GOSAT measurements are extensively validated using the ground-based measurements of $XCH_4$, obtained using the worldwide network of ground based Fourier Transform Spectrometer (FTS) called the Total

Carbon Column Observing Network (TCCON). Retrieval bias and precision of column abundance from GOSAT SWIR observations have been estimated as approximately 15-20 ppb and 1%, respectively for the NIES product using TCCON data (Morino et al., 2011; Yoshida et al., 2013). The good precision and low bias of the GOSAT instrument assures high quality of $XCH_4$ data.

### 2.2. Model simulations

Model analysis is comprised of simulations from the well-established atmospheric general circulation model (AGCM)-based chemistry-transport model (ACTM; Patra et al., 2009). The AGCM was developed by the Center for Climate System Research/National Institute for Environmental Studies/Frontier Research Center for Global Change (CCSR/NIES/FRCGC). It has been a part of the chemistry-transport model inter-comparison experiment TransCom-$CH_4$ (Patra et al., 2011) and used

in inverse modeling of $CH_4$ emissions (Patra et al., 2016). The ACTM runs at a horizontal resolution of T42 spectral truncations ($\sim 2.8^O \times 2.8^O$) with 67 sigma-pressure vertical levels. The evolution of $CH_4$ at different longitude (x), latitude (y) and altitude (z) with time in the Earth's atmosphere depends on the surface emission, chemical loss and transport and mathematically can be represented by the following continuity equation:

$$\frac{dCH_4(x,y,z,t)}{dt} = S_{CH4}(x,y,t) - L_{CH4}(x,y,z,t) - \nabla.\phi(x,y,z,t)$$

where

$CH_4$ = methane molar fraction in the atmosphere

$S_{CH4}$ = Total emissions/sinks of $CH_4$ at the surface





$L_{CH4}$ = Total loss of $CH_4$ in the atmosphere due to the chemical reactions

$\nabla . \phi$ = Transport of $CH_4$ due to the advection, convection and diffusion.


The meteorological fields of ACTM are nudged with reanalysis data from the Japan Meteorological Agency, version JRA-25 (Onogi et al., 2007). The model uses an optimized OH field (Patra et al., 2014) based on a scaled version of the seasonally varying OH field (Spivakovsky et al., 2000). The a-priori anthropogenic emissions are from Emission Database for Global Atmospheric Research (EDGAR) v4.2 FT2010 database (http://edgar.jrc.ec.europa.eu). Two different emission scenarios

(AGS and CTL) are used to examine model sensitivity to change in the underlying fluxes in simulations of the total atmospheric column and lower tropospheric column. First one is the ACTM_AGS, where all emission sectors in EDGAR42FT are kept at constant at a value for 2000, except for emissions from agriculture soils. The second one is controlled emission scenario referred by ACTM_CTL, which is based on the ensemble of the anthropogenic emissions from EDGAR32FT (as in Patra et al., 2011), wetland and biomass burning emissions from Fung et al (1991) and rice paddies

emission from Yan et al (2009). Further details about the model and these emission scenarios can be found in the previous studies (Patra et al., 2009; Patra et al., 2011; Patra et al., 2016). $XCH_4$ is calculated from the ACTM profile using the following formula.

$$XCH_4 = \Sigma_{n=2}^{60} \, CH_4 \, (n) \, * \, [(\sigma_p \, (n) + \sigma_p \, (n\text{-}1))/2 \, - \, (\sigma_p \, (n) + \sigma_p \, (n\text{+}1))/2]$$


For the first layer (n =1)

$$XCH_4 = \Sigma_{n=1} \, CH_4 \, (n) \, * \, [1\text{-} (\sigma_p \, (n) + \sigma_p \, (n\text{+}1))/2]$$

where

n = number of vertical sigma pressure layer,

$\sigma_p$ = sigma pressure level and

One simple approach i.e. the seasonal distributions of partial columnar $CH_4$ (denoted by $X_pCH_4$) at a difference of 0.2 sigma pressure layers using the same formula, is used to understand the contribution of different layers in the $XCH_4$ seasonal cycle.

Because the averaging kernels (AKs) are nearly uniform in the troposphere (Yoshida et al., 2013), this approximation does not lead to serious errors in constructing $XCH_4$ and $X_pCH_4$ in the tropical region; our study is further limited to the Indian subcontinent only. For both the CTL and AGS cases, we adjust a constant offset of 20 ppb to the modeled time series, which should make the *a priori* correction have a lesser impact on the model $XCH_4$. Because the focus of this study is seasonal and spatial variations in $XCH_4$, a constant offset adjustment should not affect the main conclusions. $XCH_4$ data are sampled at

the nearest model grid from the available observations and satellite overpass time (~ 1300 hrs LT) and then averaged over the selected partitions of the study region (Figure 2a).





## 3. Results and discussion

### 3.1 XCH$_4$ over the Indian region: View from GOSAT and ACTM simulations

This section presents an analysis of XCH$_4$ mixing ratios observed by GOSAT from Jan 2011 to Dec 2014 over the Indian region. We also use the model simulations of XCH$_4$ for same period by varying surface emissions. The total surface CH$_4$ flux optimized by the inverse analysis (Patra et al., 2016) for same period are used to elucidate the variability in XCH$_4$ mixing ratios in relation with surface fluxes. These data are averaged for three months to smooth out sporadic fluctuations and to enable the examination of seasonal variations. We characterize the seasonal mean from January to March as "Winter", April to June as "Spring", July to September as "Summer or the southwest summer monsoon" and October to December as "Autumn"; this nomenclature is maintained throughout the article. The broad features (latitudinal distributions, seasonal distributions etc.,) of simulated XCH$_4$ mixing ratios and emission fluxes for both emission scenarios (AGS and CTL) are almost similar to each other. Here we discuss the simulations and emission flux for AGS scenario only. Figure 1a-b show the observed XCH$_4$ variation for two seasons; Spring and Autumn. XCH$_4$ mixing ratios are lower during the spring seasons and higher during the autumn seasons. A strong latitudinal gradient in XCH$_4$ is observed between the Gangetic plains and remainder of India. XCH$_4$ mixing ratios show the highest value (~1880 ppb) over the IGP, eastern and northeast Indian regions. As shown in Figure 1c-d, ACTM simulations are able to reproduce the observed latitudinal and seasonal gradients; i.e., higher mixing ratios during the SW monsoon and autumn seasons and lower mixing ratios during the winter and spring seasons over the Indian region. The optimized total CH$_4$ flux shows the high emissions over the IGP region and northeast Indian regions (Figure 1e-f). Most elevated levels of XCH$_4$ are observed simultaneously with the higher emissions, suggesting a direct connection between enhanced XCH$_4$ and high surface emissions. However, this connection is not valid over all locations. For example, the emission flux has been observed as higher during the spring season than the autumn season over most of the Indian region. In contrast, higher levels of XCH$_4$ are observed during the autumn season as compared to the spring season. These inconsistencies give the hint of other factors, transport and chemistry, responsible for the XCH$_4$ distribution, apart from the emissions, over the Indian region.

To study the seasonal XCH$_4$ pattern in details, the Indian landmass was partitioned into eight regions: Northeast India (NEI), Eastern India (EI), Eastern IGP (EIGP), Western IGP (WIGP), Central India (CI), Arid India (AI), Western India (WI), Southern Peninsula (SP), and two surrounding oceanic regions, the Arabian Sea (AS) and Bay of Bengal (BOB) (Figure 2a). Regional divisions are made based on spatial patterns of emission and XCH$_4$ (Figure 1a-f), and our knowledge of seasonal meteorological conditions. Figure 2b-k shows ACTM - GOSAT comparisons of XCH$_4$ time series from Jan 2011 to Dec 2014 over the selected study regions. The climatological monthly mean of XCH$_4$ data used in Figure 2 is provided in the supplementary information (Figure S1). Observations are limited during the SW monsoon season due to GOSAT retrieval limitations under cloud cover. The model captures the salient features in the seasonal cycles at very high statistical





significance as indicated by the high correlation coefficients (r > 0.6) over the selected regions (refer to supplementary Table S1). As shown in Table S1, both tracers show, the highest correlation coefficients over SP region and cleaner oceanic regions of the AS and BOB. The high ACTM-GOSAT correlations for the low/no emission regions suggest that transport and chemistry are accurately modeled in ACTM. Although we do not have statistically significant number of observations for the

SW monsoon period, the few GOSAT data that were also simulated by ACTM over most of the study regions show high concentrations. Based on these comparisons, we can assume that model simulations can be used to understand $XCH_4$ variability over the India region. We confirmed that the modeled time series averaged over different regions with and without sampling at GOSAT sampling locations match well (r ~ 0.9).

**3.2 Seasonal cycle of $XCH_4$ and possible controlling factors**

This section discusses the average (2011-2014) $XCH_4$ annual cycle measured by GOSAT over the study regions discussed in Figure 2. The ACTM simulations with varying surface emissions optimized by the global inverse analysis (Patra et al., 2016), are further used to elucidate the seasonal variation in $XCH_4$. To investigate the role of vertical atmospheric layers in the seasonal $XCH_4$ cycle, the atmospheric column is segregated into five layers according to sigma partial pressure, starting

from the surface level ($\sigma_p = 1$) to top of the atmosphere ($\sigma_p = 0$) with an equal spacing of 0.2. The layers bounded by the boundaries between 1.0-0.8, 0.8-0.6, 0.6-0.4, 0.4-0.2, and 0.2-0.0 of sigma pressure are denoted by LT, MT1, MT2, UT and UA, respectively. The columnar $CH_4$ mixing ratios are calculated in each partial layer (denoted by $X_pCH_4$) using the same formula used for the calculations of $XCH_4$ provided in Section 2.2. The model data for the missing observations period are also used in the $XCH_4$ annual cycle to understand its complete behaviour. The climatology of optimized total $CH_4$ flux for

the same period is used to understand the link of surface emission to XCH4. Figure 3 shows the climatology of total $CH_4$ flux, climatology of $XCH_4$ and $X_pCH_4$ from model and observation over three selected regions, EIGP, SP and AI. These regions have been selected because they show distinct $XCH_4$ seasonal cycles and the distinct factors responsible for them. The remaining regions follow almost similar patterns to these three regions and hence the following discussion will equally applicable for them as well. The figures for the remaining regions are available in the supplementary information (Figures

S2, S3). Further, differences in the $X_pCH_4$, calculated at the same time as the maxima and minima of the seasonal $XCH_4$ cycle, are used to calculate the percentage contributions of respected partial columns in the seasonal amplitude of $XCH_4$ (Figure 4). All these values are estimated from ACTM simulations.

Over the EIGP region, the emission seasonality differs substantially between the CTL case and the AGS case (Figure 3g) due

to differences in emissions from wetlands, rice paddies and biomass burning; other anthropogenic emissions do not contain seasonal variations (Patra et al., 2016). The CTL case shows the emission peak in August (3.63 g $CH_4$ m$^{-2}$ month$^{-1}$), while the AGS case shows the emission peak (4.43 g $CH_4$ m$^{-2}$ month$^{-1}$) two months earlier, in June. In the AGS simulation over the EIGP region (Figure 3f), $XCH_4$ shows a peak in June that corresponds to the peak emissions (Figure 3g). However, simulated $XCH_4$ remains nearly constant until September, which is unexpected behavior based on emission seasonality. The





CTL simulation shows the peak in September, while the CTL emission scenario shows peak in August. Both simulation
        cases having different emission scenarios show peaks in September, which suggests a contribution from another factor apart
        from the emissions. Further, the $X_pCH_4$ seasonal cycle in the LT region is only partly similar to the emission pattern and the
        total column values. $X_pCH_4$ in the LT region (Figure 3e) shows an enhancement from March to June in AGS case, which
        corresponds to emissions and $XCH_4$ patterns. The $X_pCH4$ in the MT2 and UT layers show elevated mixing ratios until
September while the other layers and the emission cycle do not show such features. Hence, the upper tropospheric layers
        (MT2 and UT) contribute elevated $XCH_4$ levels from July to September over the EIGP region (Figure 3f). Further, the
        seasonal cycle amplitudes at different layers reveal that 40% of the seasonal enhancement in the observed $XCH_4$ can be
        attributed to surface emissions; only 40% of $CH_4$ is available in the lower troposphere below 600hPa, which is directly
        affected by the surface emissions (Figure 4). The remaining 60% in seasonal enhancement comes from layers above 600
hPa.

        In contrast to EIGP, a notable difference is observed in the emission seasonal cycle and $XCH_4$ seasonal cycle over SP region.
        The $XCH_4$ seasonal cycle and emission seasonal cycle are observed incompatible to each other. Both emission scenarios
        show distinct seasonal pattern; AGS shows annual high emissions from April to September, while CTL shows annual high
during August-September (Figure 3n). On the other hand, being simulated from distinct emissions scenarios having different
        seasonal cycles, the $XCH_4$ shows identical seasonal cycle corresponding to both emission scenarios: peak in October and
        broader low from May to September. This suggests that the seasonal cycle of $XCH_4$ neither follow the emission pattern, nor
        the timing of the emission peak over SP. The seasonal $X_pCH4$ cycle in the LT layer over SP shows seasonal pattern similar
        to $XCH_4$, except the peak shifts from October to November.  Surface winds from May to September over SP are from the
southern hemisphere, which effectively flushes the air with low $CH_4$ and pushes the polluted air masses from the south to the
        north India region (refer to supplementary Figure S4). Further, the distinct seasonal cycle of chemical loss is observed over
        the SP region (refer to supplementary Figure S5) compared to other study regions; the loss rate starts increasing from 6 ppb
        day$^{-1}$ in January to 12 ppb day$^{-1}$ in April, reaching a plateau from April to September (~12 ppb day$^{-1}$). These evidences
        clearly suggest that the combined effect of transport and chemistry causes the low $XCH_4$ values for the May-September
period. The peaks in the upper layers (Figure 3k-h) in October and transport from polluted continental layer in the LT layer
        (refer to supplementary Figure S4) could together contribute to the seasonal $XCH_4$ peak over SP. Over the SP region, about
        60% of the seasonal $XCH_4$ amplitude is attributed to layers below 600 hPa and remaining 40% results from the other layers
        (Figure 4). In summary, activities, dominated in the atmosphere below 600 hPa, govern most of the $XCH_4$ seasonal cycle
        over this region.


        Over the AI region, the seasonal $XCH_4$ cycle is different from those of the EIGP and SI regions. At a first glance, it seems
        the $XCH_4$ simulations (Figure 3t) follow the emission pattern over the AI region (Figure 3u). However in contrast to other
        cases mentioned above, the $X_pCH_4$ in the LT layer (Figure 3s) that is mostly affected by surface emission, does not resemble





the seasonal XCH$_4$ pattern over the AI region. The X$_p$CH$_4$ in the LT layer (Figure 4s) decreases from Jan to August and

increases until December. On the other hand, in XCH$_4$, a significant peak (~1896 ppb) is observed in August followed a decline afterward (Figure 3t). This is an outstanding example, indicating no linkage between surface emissions and XCH$_4$ in terms of seasonal peak. An enhancement in the mixing ratios of X$_p$CH$_4$ is explicitly observed from May to August in the MT2 and UT layers (Figure 3p-q) and from June to August in the UA layer (Figure 3o). In contrast to previous two regions, EIGP and SI, over the AI region, the seasonal XCH$_4$ variation in the LT and MT1 layers together contribute only about 12%

to the XCH$_4$ seasonal cycle amplitude (Figure 4). The upper layers contribute the remaining 88% (Figure 3v). Hence, based on this analysis, we conclude that instead of surface emissions, the high CH$_4$ in the upper tropospheric layers lead to the seasonal XCH$_4$ peak in August over this region. Similarly, Figure 4, Figure S2 and Figure S3 indicate that more than 60% in the seasonal amplitude of XCH$_4$ comes below 600 hPa over the regions lying in the southern half part of India, while more than 50% comes above 600 hPa over the regions mostly lying in the northern half part of India.


### 3.3 Source of higher CH$_4$ in the upper troposphere

Using ACTM simulations, we have shown that the higher CH$_4$ levels in the upper tropospheric region (~400-200 hPa) during the monsoon season contribute significantly to enhanced XCH$_4$ values over the northern regions of India. The source of higher mixing ratios in the upper troposphere as discussed in previous section can be explained by vertical transport of the

CH$_4$ emitted from the surface, because no chemical CH$_4$ source is present at this height. Figure 5a-d shows the latitudinal cross section of the convective transport rate (in ppb day$^{-1}$) along with height and vertical velocity (hPa s$^{-1}$) averaged over 83-93°E for different seasons in 2011 (the ACTM_AGS simulation case). The positive/negative values of convective transport rate and vertical velocity in Figure 5a-d indicate the gain/loss of mass and downward/upward motions, respectively. Rapid updrafts, as indicated by higher negative values of vertical velocity, of higher CH$_4$ surface emissions by deep

convection during the monsoon season are aided by the local topography over the IGP region (north of 20°N and east of 79°E in the Indian region). These updrafts cause higher mixing ratios of CH$_4$ in the upper tropospheric region (Figure 5g). The surface CH$_4$ mixing ratios are dissipated at an average rate of ~10 ppb day$^{-1}$ during Spring-Autumn seasons (Figure 5b-d), and accumulate in the upper troposphere height at a similar rate; peak accumulation height varies with season. The horizontal cross-section of CH$_4$ at 200 hPa and wind vectors is plotted in Figures 5i-l for understanding the spatial extent of

uplifted CH$_4$-rich air over the Indian region. The CH$_4$-rich air mass in the upper troposphere (~200 hPa) is further encountered with the anticyclonic winds during the SW monsoon season, which trap CH$_4$ and leads to widespread enhancement covering South Asia, and extending through the East Asia (Figure 5k). As a result of this, higher levels of CH$_4$ at upper troposphere, are not only limited over the regions where intense surface sources exist, the regions where surface sources are not comparatively strong are also covered by the high levels of CH$_4$ and hence contribute in the seasonal peak of

XCH$_4$. After the SW monsoon season, the high westerly jet breaks the upper tropospheric anticyclone and elevated levels shift southward (Figure 5l) and cause higher CH$_4$ mixing ratios over southern India during the autumn season. Thus the convective updraft of high-CH$_4$ air mass, followed by horizontal spreading over the larger area by anticyclonic circulation,





control the redistribution of high CH$_4$ concentrations over the upper tropospheric region of northern part of India during SW monsoon season and over southern peninsula at this height during the early autumn season.


Like in Section 3.2, previous studies also show a shift in XCH$_4$ seasonal peak from August to October as we move from northern to Southern India and the major cause was discussed on the basis of emission from two major agricultural seasons of Kharif (May to October) and Rabi (November to April) (Hayashida et al., 2013; Kavitha et al., 2016). However, the detailed analysis of emission and transport component suggests that apart from surface emissions, the shift in XCH$_4$ seasonal

peak could explained by shifting of high CH$_4$ levels associated with the anticyclonic winds at upper tropospheric height from northern to southern Indian region, respectively.

### 4. Conclusions

The dry-air mole fractions of methane (XCH$_4$) measured by GHGs Observation SATellite (GOSAT) were closely analyzed

here over India and the surrounding seas. The region of interest is divided in to 8 sub-regions, namely, Northeast India (NEI), Eastern India (EI), Eastern IGP (EIGP), Western IGP (WIGP), Central India (CI), Arid India (AI), Western India (WI), Southern Peninsula (SP), and two surrounding oceanic regions, the Arabian Sea (AS) and Bay of Bengal (BOB. The JAMSTEC's atmospheric chemistry-transport model (ACTM) of CH$_4$ and total surface flux optimized by the inverse analysis are used for bridging the transport and emission information to observed XCH$_4$ mixing ratios and address their roles

in the annual cycle in detail. We have been observed that distinct spatial and temporal features of XCH$_4$ are not only governed by the heterogeneous surface emissions, but also due to complex atmospheric transport mechanisms caused by the seasonally varying monsoon. The seasonal XCH$_4$ patterns often show a fair correlation between emissions and XCH$_4$ over the regions residing in the northern half of India (north of 15$^o$N: NEI, EI, EIGP, WIGP, CI, WI, AI), which implies XCH$_4$ levels are closely associated with the distribution of emission sources. However, detailed analysis of transport and emission

reveal that only less than 40% of seasonal enhancement in the observed XCH$_4$ can be attributed to surface emissions over these regions except AI, as only this amount of CH$_4$ enhancement is available in the lower troposphere (below 600 hPa), which is directly affected by the surface emissions. In fact, ~40-60% of the CH$_4$ enhancement is in the uplifted air mass between 600-200 hPa over these regions. In contrast, over semi-arid AI region, as much as ~88% contributions to the XCH$_4$ seasonal cycle amplitude come from above 600 hPa, and only ~12% are contributed by the atmosphere below 600 hPa. The

primary cause of the higher contributions from above 600 hPa over the northern Indian region is the characteristic transportation mechanisms in the Asian monsoon regions. The persistent deep convection during the southwest (SW) monsoon season (June-August) causes strong updrafts of CH$_4$ from the surface to upper troposphere, which is then distributed by anticyclonic winds over the northern Indian region. These transport mechanisms caused the elevated CH$_4$ mixing ratios in the upper troposphere hence contributed significantly to the seasonal peak in XCH$_4$ over northern India. In

contrast to these regions, over the SP region, the major contributions (about 60%) to XCH$_4$ seasonal amplitude come from the lower atmosphere (~1000-600 hPa). Both transport and chemistry dominate in the lower atmosphere over SP region and,





as a result of it, the seasonal variation of $XCH_4$ is not corresponding to the seasonality of the local emissions. As upper level anticyclones do not cover the southern Indian region in its active phase during the SW monsoon season, the enhancement in $XCH_4$ is not observed over the southern peninsular region during the SW monsoon season.


Most satellite sensors are designed to provide total columnar observations of atmospheric chemical species. This study opens a new window for interpreting columnar measurements for surface emissions of greenhouse gases particularly over the Asian monsoon region where characteristic meteorology dominates, and should aid users in carefully applying scientific data in the future to not draw erroneous conclusions.


**Acknowledgements**

The Environment Research and Technology Development Fund (A2-1502) of the Ministry of the Environment, Japan, supported this research. The data used for preparing the figures, and table could be available on request. The corresponding author may be contacted for the same.

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





**Figures.**

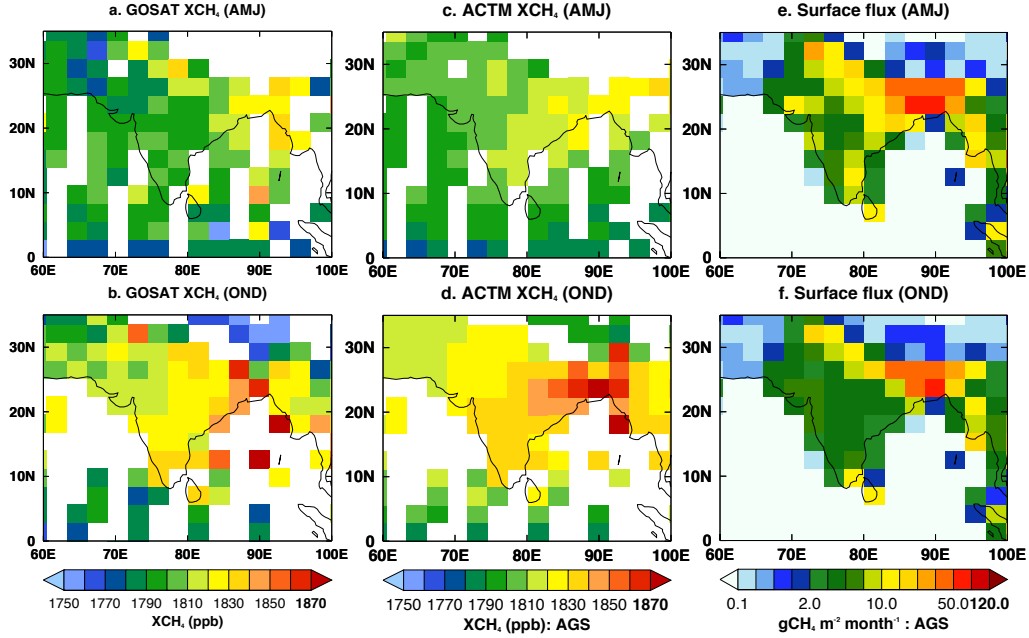


**Figure 1:** Average seasonal distributions (from 2011 to 2014) of XCH$_4$ obtained from GOSAT observations (a-b), ACTM simulations (c-d) and CH$_4$ emission consisting of all the natural and anthropogenic emissions (e-f: ACTM_AGS case) over the Indian region. Optimized emissions are shown from a global inversion of surface CH$_4$ concentrations (Patra et al., 2016) and multiplied by a constant factor of 12 for a clear visualization. The ACTM is first sampled at the location and time of

GOSAT observations and then seasonally averaged. The white spaces in panels (a-d) are due to the missing data caused by satellite retrieval limitations from cloud cover.





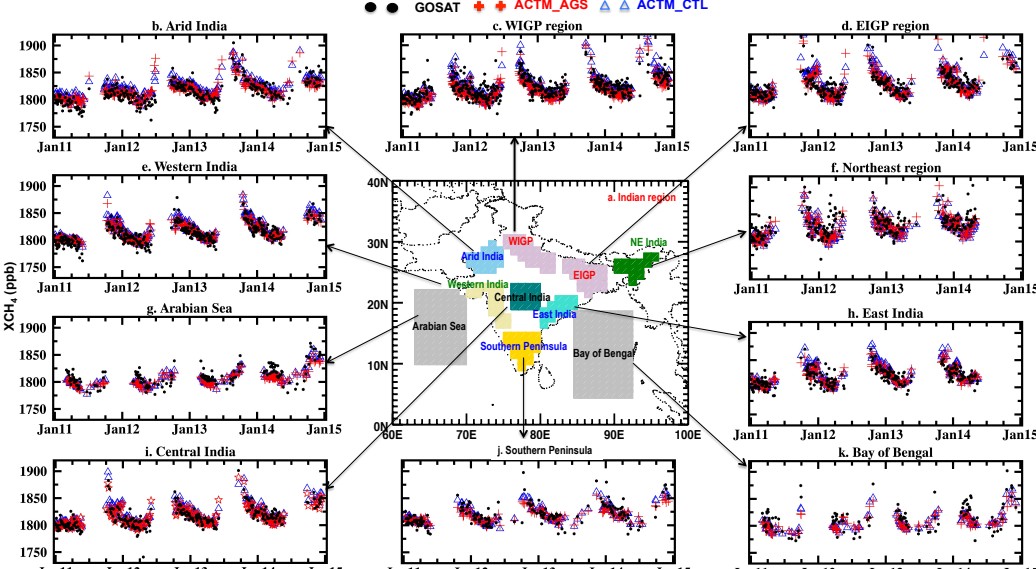

Figure 2: (a) The map of the regional divisions (shaded) for the time series analysis. (b-l) Time series of XCH₄ over the

selected regions (shown in map) as obtained from GOSAT and simulated by ACTM for two different emission scenarios, namely, ACTM_AGS and ACTM_CTL. The gaps are due to the missing observational data.




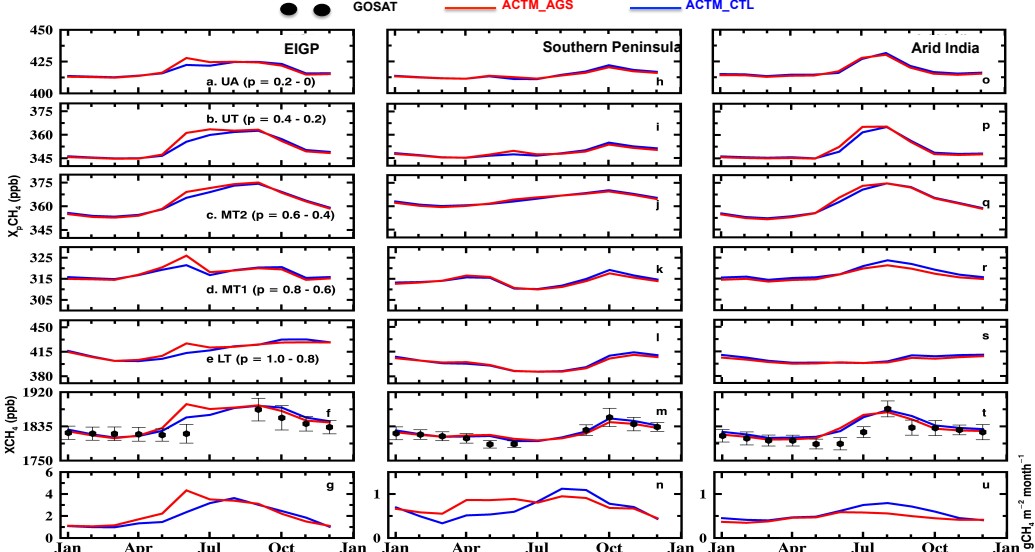

Figure 3: The bottom panels show the monthly mean climatology of the total optimized CH$_4$ emissions (panels g, n, u); estimated after performing the global inverse analysis (Patra et al., 2016). The second bottom panels show XCH$_4$ obtained from the GOSAT observations (black circles in panels f, m, t) and ACTM simulations (panels f, m, t) over the Eastern IGP (first column), Southern Peninsula (second column) and Arid India region (third column). Monthly climatology is based on the monthly mean values for the period of 2011-2014 for all the values. The error bars in the GOSAT monthly mean values depict the 1-sigma standard deviations for the corresponding months (f, m, t). The 1-sigma values are not plotted for the model simulations to maintain figure clarity. Simulations are based on two different emission scenarios namely ACTM_CTL (blue lines) and ACTM_AGS (red lines) based on the different combinations of emissions. The upper five panels show the monthly climatology of partial columnar methane (denoted by x$_p$CH$_4$) calculated at five different partial sigma-pressure layers; 1.0-0.8 (e, l, s), 0.8-0.6 (d, k, r), 0.6-0.4 (c, j, q), 0.4-0.2 (b, I, p) and 0.2-0.0 (a, h, n). Please note that the y scales in the emission plots over southern peninsula and Arid India (n and u) are different than over the EIGP region (g).





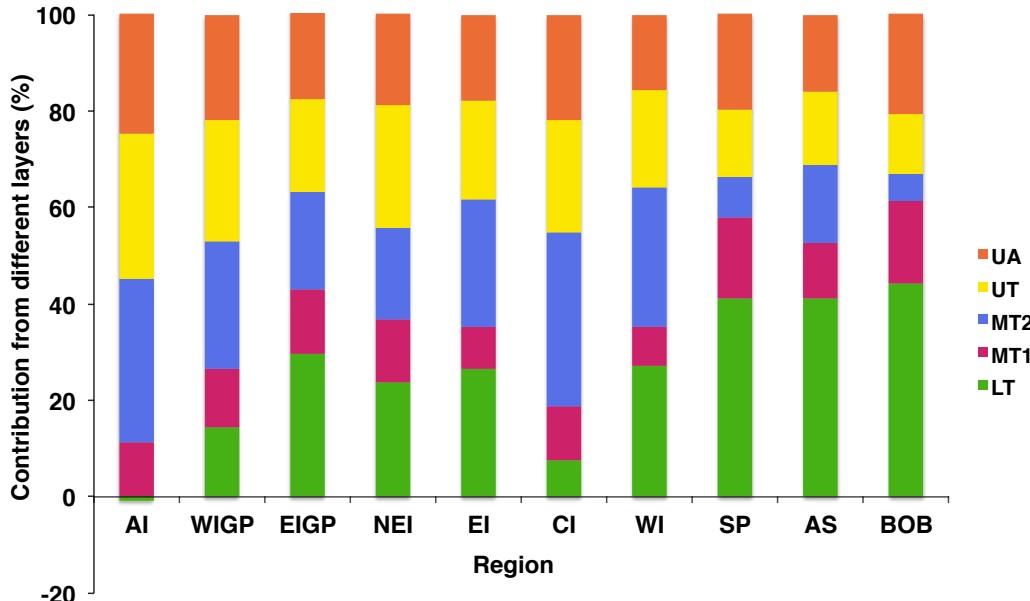

Figure 4: Contributions of partial columns in the seasonal amplitude of XCH$_4$ over selected regions for AGS case. Differences in the X$_p$CH$_4$, calculated at the same time as the maxima and minima of the seasonal XCH$_4$ cycle, are used to calculate the percentage contributions of respected partial columns in the seasonal amplitude of XCH$_4$.





**Figure 5:** Vertical structure of seasonally averaged CH$_4$ transport rate due to the convection (a-d, in ppb day$^{-1}$) and CH$_4$ mixing ratios (e-h from AGS scenarios) averaged over 83-93°E for the year of 2011. Positive and negative transport rate





values represent the accumulation and dissipation of mass, respectively. The contour lines in the fist (a-d) and second (e-h) columns depict the average omega velocity (in hPa s$^{-1}$) and u wind component, respectively for the same period. The solid contour lines show the positive values and dotted lines show negative values. Positive and negative values of the omega

520  velocity represent downward and upward motions, respectively. The zero value of u wind shows wind is pure either southerly or northerly. White spaces in zonal-mean plots (a- h) show the missing data due to orography. The rightmost column depicts the maps of averaged CH$_4$ and wind vectors (in m s$^{-1}$; arrow) during all the four seasons in 2011 at 200 hPa height (i-l).