# Peer review of "What controls the seasonal cycle of columnar methane observed by GOSAT over different regions in India."

_Atmospheric Chemistry and Physics, 2017_

## Referee Comment (RC1) · Anonymous Referee #1 · 11 May 2017

This paper shows that variability in XCH4 measured by satellite over India cannot be simply attributed to spatial/seasonal variability of local sources but reflects also major influences from the free troposphere. This is a simple argument and is of some interest because there is indeed temptation to make such an attribution, and the paper shows clearly that it is incorrect. The paper also presents a nice analysis of the factors controlling methane in the free troposphere over India. The scope of the paper is limited (hence my rating of "Fair") because it applies only to India, and because a more thorough analysis of XCH4 data (such as with an inversion) would obviate the need for the incriminated assumption. I was hoping to get a better understanding of methane sources in India but in fact this is not what the paper is about. Still, one could cite the

paper as an admonition to pay attention to the free troposphere when making simple interpretation of XCH4.

The presentation of the paper could be improved to make it more attractive: the writing is fastidious with too much details, the grammar and style are often poor, the postage-stamp figures scare the reader away (could you make do with fewer panels), and the math isn't clean.

One minor thing: on line 42, delete "increase in".

---

## Referee Comment (RC2) · Anonymous Referee #2 · 30 Jun 2017

**General comments**

The manuscript discusses the vertical distribution of tropospheric methane over South Asia based on ACTM calculations and GOSAT data. It is a valuable contribution for our understanding of transport and emission contributions to methane mixing ratios at different altitudes, in particular with regard to the influence of convection during the south westerly summer monsoon.

[Figure]

**Specific comments**

The text is rather unstructured in some parts, such as subsections 3.1, 3.2 or the conclusion sections) which makes it difficult to read. Having more paragraphs and using less abbreviations could easily improve this. Several times abbreviations are introduced which are not needed because the term does not get used frequently in the text (e.g. first line in abstract SLCF). The abbreviations used for the regions and for the pressure levels are not very intuitive. Abbreviations for the pressure levels may not be needed at all.

The abstract is rather long and too detailed.

The text uses sigma pressure level and pressure numbers in hPa interchangeably. This should be made consistent.

I was wondering whether you checked if for the years studied here, namely 2011–2014, it was checked if the southwest monsoon fitted into the season scheme that was used here, i. e. was June truly a pre-monsoonal spring month in all four years and did the southwest monsoon prevail through September.

**Line 86 f** Here a contrast to the GOSAT TIR data is mentioned. This is not at all connected to the previous paragraph plus it has not yet been clearly stated that SWIR data is used in this manuscript. It does become later, though, but here the mentioning of the TIR data is confusing.

**Line 140** What does AGS stand for?

**Line 158 f.** It becomes obvious later in section 3.2. what you mean here, but on first reading it was totally unclear to me.

**Line 165** What is the maximum spatial difference that will occur?

**Line 198 f.** I don't understand what was done here. How was the climatology in Figure S1 used for the data shown in Figure 2? Shouldn't it be the other way round, that Figure S1 shows the climatological means resulting from the timeseries shown in Figure 2?

**Section 3.1** The discussion lacks a comparison between the GOSAT data and the ACTM results. In particular in June for EIGP, you discuss the difference in emission between the AGS and the CTL model run, but for both scenarios agreement with GOSAT looks rather poor compared to the other months. For arid India this holds also for July. This disagreement makes it difficult to discuss the difference between the model runs in great detail as none of them seem to reproduce the measurements very well.

Also, the monthly climatologies shown in Figure S1 have gaps that result from data lacking due to cloud cover. However, seasonalities shown in Figure 3, although based on the climatological means, do not who these gaps for the model data. So, was the treatment of the model data somehow different for this part of the study than for the previous one? I guess this is what you mean in **Line 208** by 'without sampling'?

**Wording**

**Line 54** 'which could fill in gap' does not make sense.

**Line 60 f.** 'The Indo-Gangetic Plane ... Himalayas' — The wording here does not make sense.

**Line 68** It's either convection or convective uplift but not convection uplift.

**Line 71** Typo in 'plateau'.

**Line 77** 'related the high XCH4 values correspond' does not make sense.

**Line 81** What do you mean by 'inferring the local emissions **to** the higher emissions' ?

**Line 86 f** Grammar in this sentence is not logical.

**Line 92 f** 'under the limitations of satellite' — Please re-formulate.

**Line 142** change 'kept at constant at a value' to 'kept at a constant value'

**Line 156** Why is there an 'and' here?

**Line 223/224** 'will equally applicable' does not make sense.

**Line 320** 'we have been observed' does not make sense.

**Line 333/334** Grammar is not logical here

**Figures and Tables**

In general, most figures are too small and use too small fonts for labels and annotations, in particular axis labels. Panel labels a, b, c ... are very difficult to spot which makes it complicate to follow the discussion in the text.

**Figure 2** This figure has too many panels resulting in them being too small. Think about separating the map into a decently sized figure on its own and presenting panels b–l in a classical two-column scheme with larger panels.

**Figure 5** Y-axis has no label and no units.

**Figure 5** Caption: change 'year of 2011' to 'year 2011'.

**Table S1** Table says 'South India' but 'Southern Peninsula' is used throughout the text. Think about also including the abbreviations for the regions used in the text into the table.

[Figure]

---

## Author Comment (AC1) · 3 Jul 2017

Please find the response to reviewer's comments and revised manuscript in the Supplement zip file .

Please also note the supplement to this comment:
https://www.atmos-chem-phys-discuss.net/acp-2017-344/acp-2017-344-AC1-supplement.zip

---

## Author Response (AR1)

**Reply to Anonymous Referee #1 ()**

We thank the reviewer for providing us with feedback on the manuscript quickly. We have attempted to revise the manuscript based on the comments. Hope the readability of the manuscript is now improved.

This paper shows that variability in XCH4 measured by satellite over India cannot be simply attributed to spatial/seasonal variability of local sources but reflects also major influences from the free troposphere. This is a simple argument and is of some interest because there is indeed temptation to make such an attribution, and the paper shows clearly that it is incorrect. The paper also presents a nice analysis of the factors controlling methane in the free troposphere over India. The scope of the paper is limited (hence my rating of "Fair") because it applies only to India, and because a more thorough analysis of XCH4 data (such as with an inversion) would obviate the need for the incriminated assumption. I was hoping to get a better understanding of methane sources in India but in fact this is not what the paper is about. Still, one could cite the paper as an admonition to pay attention to the free troposphere when making simple interpretation of XCH4.

Response: The Indian region (South Asia in general) exerts a significant impact on the global $CH_4$ emissions. About 10% of total $CH_4$ emissions (550 Tg/yr) is emitted from the South Asian region (Patra et al., 2013). Investigations of sectoral emission of $CH_4$ over Indian region are thus significantly important, particularly over the Indo-Gangetic Plain (IGP) as the region is well-known hotspot of emissions of anthropogenic greenhouse gases. The transport mechanisms over the IGP and the Himalayan region are of global importance for transport and transformation of methane and other air pollutants; refer for example the aim of the SPARC/IGAC jointly sponsored activity - Atmospheric Composition and the Asian Monsoon (ACAM) project (https://www2.acom.ucar.edu/acam). It may also be pointed out here that there are several important projects being planned to achieve the ACAM goals (e.g., Stratospheric and upper tropospheric processes for better climate predictions; http://www.stratoclim.org/). Thus, we believe understanding the transport of $CH_4$ in one of the strongest monsoonal regions of the globe is not likely to of a limited interest.

The space-based observations are limited to total columnar methane mixing ratios ($XCH_4$). However, our knowledge of handling total column data in an inverse modelling system is limited and serious systematic biases require attention (e.g., Ostler et al., AMT, 2017). Thus, it is important to understand the source receptor relationships before inverse modeling of regional sources and sinks. Linking the surface emissions to the $XCH_4$ observations over the Indian region is not very straightforward because of the coexistence of deep convection and large emissions of $CH_4$ from a variety of both natural and anthropogenic sources. Therefore, recognizing the role of transport is extremely important in order to understand the contributions of emission signals to the $XCH_4$ variabilties.

We have revised manuscript text to clarify the aims of this manuscript in a significantly revised version. We would like to request the reviewer to take a look in to the revised version now uploaded.

The presentation of the paper could be improved to make it more attractive: the writing is fastidious with too much details, the grammar and style are often poor, the postage stamp figures scare the reader away (could you make do with fewer panels), and the math isn't clean.

Response: We apologize for the sloppiness in writing the submitted version. We have put our best effort in improving the presentation of text. We add or remove some unwanted and confusing text throughout the manuscript. The presentation of results and conclusion become now more clear and straightforward. We present the column calculation equation in Section 2.2 now in a simplistic way. A language editing service checked a preliminary version, but some late revisions did not go through the language check. Thank you very much for kindly providing us with quick comments.

We have improved presentations of all figures after readjusting and modifying the text fonts, axis label and titles etc as per the comments from both reviewers. Given below are some of the specifics:

Figure 1: We have changed labels and title font size in this revised figure.

Figure 2: We have also changed labels and title font size in this figure. We have removed the x-axis tick labels from top three rows, changed y-axis scale from 1760 to 1920 ppb, moved panel titles inside the panels and used that extra space to increase the panel size.

Figure 3: We have increased the font and label size, decoupled layer information from each panel, shifted this information to the rightmost part of panels, placed panel numbers at the top left corner of each panel, moved the stations name to the top side of plot and changed the aspect ratio of the revised figure.

Figure 4: We have increased the font and label size of this revised figure.

Figure 5: We have changed the fonts of all labeling, moved the panel title into the plot area with shaded (white) background, increased the length of tick marks, used the text "Vertical cross-sections" for left and middle column and "Horizontal cross section" for rightmost column on the top of figure, changed the color scale (new scale: 1750 – 1930 ppb) and shift the season names to the rightmost part of revised figure.

One minor thing: on line 42, delete "increase in".
Response: The sentence has modified now in the revised manuscript.

References:

Patra, P. K., Canadell, J. G., Houghton, R. A., Piao, S. L., Oh, N.-H., Ciais, P., Manjunath, K. R., Chhabra, A., Wang, T., Bhattacharya, T., Bousquet, P., Hartman, J., Ito, A., Mayorga, E., Niwa, Y., Raymond, P. A., Sarma, V. V. S. S., and Lasco, R.: The carbon budget of South Asia, Biogeosciences, 10, 513-527, doi:10.5194/bg-10-513-2013, 2013.

Ostler, A., Sussmann, R., Patra, P. K., Houweling, S., De Bruine, M., Stiller, G. P., Haenel, F. J., Plieninger, J., Bousquet, P., Yin, Y., Saunois, M., Walker, K. A., Deutscher, N. M., Griffith, D. W. T., Blumenstock, T., Hase, F., Warneke, T., Wang, Z., Kivi, R., and Robinson, J.: Evaluation of column-averaged methane in models and TCCON with a focus on the stratosphere, Atmos. Meas. Tech., 9, 4843-4859, doi:10.5194/amt-9-4843-2016, 2016.

**Reply to Anonymous Referee #2 ()**

General comments:
The manuscript discusses the vertical distribution of tropospheric methane over South Asia based on ACTM calculations and GOSAT data. It is a valuable contribution for our understanding of transport and emission contributions to methane mixing ratios at different altitudes, in particular with regard to the influence of convection during the southwesterly summer monsoon.

Response: We thank the reviewer for careful evaluation and providing us feedback on the manuscript. We have revised the manuscript based on the general and specific comments from both of the reviewers. We have also worked very carefully on improving clarity of the manuscript text and figures. We hope the revised version is easy to follow the results and discussion of this study.

Specific comments
The text is rather unstructured in some parts, such as subsections 3.1, 3.2 or the Conclusion sections) which makes it difficult to read. Having more paragraphs and using less abbreviation could easily improve this. Several times abbreviations are introduced which are not needed because the term does not get used frequently in the text (e.g. first line in abstract SLCF). The abbreviations used for the regions and for the pressure levels are not very intuitive. Abbreviations for the pressure levels may not be needed at all. The abstract is rather long and too detailed.

Response: We have revised the whole manuscript in accordance with your suggestions and the comments from Reviewer #1. The whole manuscript text has been revised significantly to the best of our ability. The abstract is made concise and straightforward. Hopefully, the meaning of the text is clearer and straightforward now.

We tried to avoid the use of abbreviations as much as possible. As we stated in the text, the atmospheric column was segregated into five sigma-pressure ($\sigma_p$) layers with an equal spacing of 0.2 starting from the surface level ($\sigma_p=1$), corresponding to Lower Troposphere (LT), Mid-Troposphere1 (MT1), Mid-Troposphere2 (MT2), Upper Troposphere (UT) and Upper Atmosphere (UA), respectively. Those names are explicitly stated in the revised manuscript. To avoid using long names of those layers, we keep the abbreviations in the discussion.

The text uses sigma pressure level and pressure numbers in hPa interchangeably. This should be made consistent.
Response: The sigma pressure coordinate was used only to divide the atmospheric column into partial columns to avoid the effect of topography in the partial column calculations. In the later part on discussion, we used pressure numbers only.

I was wondering whether you checked if for the years studied here, namely 2011–2014, it was checked if the southwest monsoon fitted into the season scheme that was used here, i.e. was June truly a pre-monsoonal spring month in all four years and did the southwest monsoon prevail through September.

Response: Our main focus in this study is the analysis of mean features in XCH$_4$ seasonal cycle. To show the seasonal cycle clearly, we divided a year into four periods of 3 months duration, which is commonly used in meteorological studies (e.g., Rao, 1976). We repeated the analysis including June and excluding September from the southwest monsoon period, but didn't find significant difference that could affect our conclusion. To investigate the year-to-year variability of the period of the southwest monsoon season, we also individually plotted these transport processes for each year from 2011 to 2014 and didn't find any significant inter-annual variability in their nature. Please look at the following figure for reference. We have added this clarification in the main manuscript also.

[Figure]

Line 86 f

Here a contrast to the GOSAT TIR data is mentioned. This is not at all connected to the previous paragraph plus it has not yet been clearly stated that SWIR data is used in this manuscript. It does become later, though, but here the mentioning of the TIR data is confusing.

Response: Here the purpose of mentioning TIR band is just to provide information to readers about the availability of vertical $CH_4$ data, although they can't be used due to their limited validity in this study. This is the reason why we have to use the model simulations instead of the observation. But to avoid confusion we have removed the sentence in the revised manuscript.

Line 140
What does AGS stand for?
Response: Sorry for this confusion. AGS is not an abbreviation. AGS is a name of the ensemble emission dataset that is used in the ACTM as an a priori emission case. In this case the EDGAR4.2FT emissions from agricultural sectors are only allowed to change during the period of inversion (2001-2013), while all other anthropogenic emissions are kept constant at the 2002 level. Now we have modified the sentence in the following manner to make it clearer.

"The model sensitivity for emission is examined by two cases of emission scenarios based on different combination of sectoral emissions. First one is referred to the 'AGS', where all emission sectors in EDGAR42FT are kept at a constant value for 2000, except for emissions from agriculture soils. The second one is controlled emission scenario referred to 'CTL', which is based on the ensemble of the anthropogenic emissions from EDGAR32FT (as in Patra et al., 2011a), wetland and biomass burning emissions from Fung et al. (1991) and rice paddies emission from Yan et al. (2009)".

Line 158 f.
It becomes obvious later in section 3.2. what you mean here, but on first reading it was totally unclear to me.
Response: We have removed this sentence from Section 2.2 and add a sentence in Section 3.2 in the revised manuscript to avoid the confusion.

"The partial columnar $CH_4$ are calculated within different $\sigma_p$ layers (denoted by $X_pCH_4$) using the same formula for $XCH_4$, as in Section 2.2."

Line 165
What is the maximum spatial difference that will occur?
Response: The maximum spatial difference occurs about 1.2º.

Line 198 f.

I don't understand what was done here. How was the climatology in Figure S1 used for the data shown in Figure 2? Shouldn't it be the other way round, which Figure S1 shows the climatological means resulting from the time series shown in Figure 2?

Response: Sorry for the confusing sentence. Please read our modified sentence (lines 217-218) as:

"The monthly mean climatology, resulting from the time series shown in Figure 2, is shown in the supplementary information (Figure S1)".

Section 3.1
The discussion lacks a comparison between the GOSAT data and the ACTM results. In particular in June for EIGP, you discuss the difference in emission between the AGS and the CTL model run, but for both scenarios agreement with GOSAT looks rather poor compared to the other months. For arid India this holds also for July. This disagreement makes it difficult to discuss the difference between the model runs in great detail as none of them seem to reproduce the measurements very well.

Response: In Figure 3, there is a mismatch between ACTM emission scenarios and GOSAT observations in June for EIGP and also in some months for Arid India region. One possible reason is that there are not sufficient observations (less than 10 data) involved in the monthly mean corresponding to those months due to GOSAT sparseness in the coverage. However, despite the lack of data number in monsoon season, we can find both model scenarios are generally able to capture some of the available observations during the monsoon months in Figure 2. Here our main focus in comparison of GOSAT with ACTM is to confirm the ability of the model in producing the seasonal phase of $XCH_4$. For these reasons, we do not discuss the discrepancy between GOSAT and ACTM in detail here.

Also, the monthly climatologies shown in Figure S1 have gaps that result from data lacking due to cloud cover. However, seasonalities shown in Figure 3, although based on the climatological means, do not who these gaps for the model data. So, was the treatment of the model data somehow different for this part of the study than for the previous one? I guess this is what you mean in Line 208 by 'without sampling'?

Response: In the revised manuscript we clearly stated that "climatology" means the monthly mean values for the period of 2011-2014 (see figure captions of Figure 3 and Figure S1 in the revised manuscript). Figure S1 shows the climatological monthly mean of all of the available GOSAT data. On the other hand, in Figure 1 and Figure 2, we have sampled the model data that are collocated and coincident with GOSAT observations. In Figure 3, we included all of the model data in the panels of three selected regions (a6, b6, and c6) to show the seasonal cycle clearly.

Wording

Line 54
'which could fill in gap' does not make sense.
Response: We have modified the sentence to:
"Recent technological advances have made it possible to detect spatial and temporal variations in atmospheric $CH_4$ from space (Frankenberg et al., 2008; Kuze et al., 2009), which could provide global and dense data over the regions uncovered by ground, aircraft and ship-based measurements, albeit at a lower accuracy than the *in situ* measurements.'

Line 60 f.
'The Indo-Gangetic Plane ... Himalayas' — The wording here does not make sense.

Response: We have modified this sentence in the revised manuscript.
"The Indo-Gangetic Plain (IGP) located in the foothills of the Himalayas is one of the most polluted regions in the world, which hosts 70% of coal-fired thermal power plants in India and experiences intense agricultural activity".

Line 68
It's either convection or convective uplift but not convection uplift.
Response: We have modified this sentence in the revised manuscript.
" Rainfall during the SW monsoon season cause higher $CH_4$ emissions from the paddy fields and wetlands while the persistent deep convection results the updraft of $CH_4$-laden air mass from the surface to the upper troposphere during the same season, which is then confined by anticyclonic winds at the this height."

Line 71
Typo in 'plateau'.
Response: Corrected.

Line 77
'related the high XCH4 values correspond' does not make sense.

Response: We have modified the sentence to:
"Previous studies have linked these high XCH4 levels to the strong surface $CH_4$ emissions particularly from the rice cultivation over the Indian region, because they showed statistically significant correlations over certain regions."

Line 81
What do you mean by 'inferring the local emissions to the higher emissions'?
Response: Here we want to mention that high XCH4 cannot be directly linked with high local/regional emissions. We have changed the sentence for better clarity to:

"However, inferring local emissions directly from variations in $XCH_4$ is ambiguous particularly over the Indian regions under the influence of monsoon meteorology, because $XCH_4$ involves contributions of $CH_4$ abundances from all altitudes along the solar light path."

Line 86 f
Grammar in this sentence is not logical.
Response: This sentence has been removed from the revised version as per your suggestion.

Line 92 f
'under the limitations of satellite' — Please re-formulate.
Response: This sentence has been removed from the revised version.

Line 142
change 'kept at constant at a value' to 'kept at a constant value'
Response: Changed

Line 156
Why is there an 'and' here?
Response: Sorry for this typo error. We have removed 'and' in the revised manuscript.

Line 223/224
'will equally applicable' does not make sense.
Response: This sentence is eliminated in the revised version.

Line 320
'we have been observed' does not make sense.
Response: We have modified the sentence in the revised manuscript.
"We have shown that the distinct spatial and temporal variations of $XCH_4$ observed by GOSAT are not only governed by the heterogeneity in surface emissions but also due to complex atmospheric transport mechanisms caused by the seasonally varying Asian monsoon."

Line 333/334
Grammar is not logical here
Response: We have modified the sentence in the revised manuscript.
"The persistent deep convection during the southwest monsoon season (June-August) causes strong updrafts of $CH_4$-rich air mass from the surface to upper tropospheric heights (~200 hPa), which is then confined by anticyclonic winds at this height. The anticyclonic confinement of surface emission over a wider South Asia region leads to strong contribution of the upper troposphere in formation of the $XCH_4$ peak over most regions in northern India, including the semi-arid regions with extremely low $CH_4$ emissions."

Figures and Tables
In general, most figures are too small and use too small fonts for labels and annotations, in particular axis labels. Panel labels a, b, c ... are very difficult to spot, which makes it complicate to follow the discussion in the text.
Response: We apologize for this. We have improved presentations of all figures after readjusting and modifying the text fonts, axis label and titles etc. Given below are some of the specifics:
Figure 1: We have changed labels and title font size in this revised figure.

Figure 2: We have also changed labels and title font size in this figure. We have removed the x-axis tick labels from top three rows, changed y-axis scale from 1760 to 1920 ppb, moved panel titles inside the panels and used that extra space to increase the panel size.

Figure 3: We have increased the font and label size, decoupled layer information from each panel, shifted this information to the rightmost part of panels, placed panel numbers at the top left corner of each panel, moved the stations name to the top side of plot and changed the aspect ratio of the revised figure.

Figure 4:  We have increased the font and label size of this revised figure.

Figure 5: We have changed the fonts of all labeling, moved the panel title into the plot area with shaded (white) background, increased the length of tick marks, used the text "Vertical cross-sections" for left and middle column and "Horizontal cross section" for rightmost column on the top of figure, changed the color scale (new scale: 1750 – 1930 ppb) and shift the season names to the rightmost part of revised figure.

Figure 2
This figure has too many panels resulting in them being too small. Think about separating the map into a decently sized figure on its own and presenting panels b–l in a classical two-column scheme with larger panels.
Response: We increased the area within each panels after readjusting the fonts of labels and moving the panel title within the plot area. We want to show the regions in India map and XCH4 variability over those regions together and thus we prefer to keep the arrangement of the panels as before. We think this arrangement will be convenient for the reader to track the location and XCH4 distribution in the same figure.

Figure 5
Y-axis has no label and no units.
Response. We have added the label for Y-axis.

Figure 5
Caption: change 'year of 2011' to 'year 2011'.
Response: Changed

Table S1

Table says 'South India' but 'Southern Peninsula' is used throughout the text. Think about also including the abbreviations for the regions used in the text into the table.

Response: Changed

[revised manuscript text omitted]

Naveen Negi 7/3/2017 10:42 AM

林田佐智子 6/21/2017 4:16 PM

Naveen Negi 7/3/2017 11:03 AM

林田佐智子 6/21/2017 4:26 PM

Prabir Patra 6/19/2017 12:25 PM

Naveen Negi 7/3/2017 11:05 AM

Prabir Patra 6/19/2017 12:27 PM

Naveen Negi 7/31/2017 4:07 PM

Prabir Patra 6/19/2017 12:35 PM

Naveen Negi 7/3/2017 11:07 AM

Naveen Negi 8/14/2017 1:11 PM

Prabir Patra 6/19/2017 12:41 PM

Naveen Negi 8/14/2017 1:12 PM

Prabir Patra 6/19/2017 12:42 PM

Naveen Negi 7/3/2017 11:29 AM

林田佐智子 6/21/2017 4:41 PM

Prabir Patra 6/21/2017 10:19 PM

Naveen Negi 6/22/2017 11:13 AM

林田佐智子 6/21/2017 4:50 PM

Naveen Negi 7/3/2017 11:30 AM

Prabir Patra 6/19/2017 12:47 PM

Prabir Patra 6/19/2017 1:43 PM

Naveen Negi 7/31/2017 4:08 PM

Prabir Patra 6/19/2017 1:44 PM

[revised manuscript text omitted]